# Recent Progress of Cu-Catalyzed Azide-Alkyne Cycloaddition Reactions (CuAAC) in Sustainable Solvents: Glycerol, Deep Eutectic Solvents, and Aqueous Media

**DOI:** 10.3390/molecules25092015

**Published:** 2020-04-26

**Authors:** Noel Nebra, Joaquín García-Álvarez

**Affiliations:** 1UPS, CNRS, LHFA UMR 5069, Université de Toulouse, 118 Route de Narbonne, 31062 Toulouse, France; 2Departamento de Química Orgánica e Inorgánica, Instituto Universitario de Química Organometálica “Enrique Moles” (IUQOEM), Facultad de Química, Universidad de Oviedo, E-33071 Oviedo, Spain

**Keywords:** CuAAC, water, glycerol, deep eutectic solvents, cycloadditions, green chemistry, metal-catalysis

## Abstract

This mini-review presents a general overview of the progress achieved during the last decade on the amalgamation of CuAAC processes (copper-catalyzed azide-alkyne cycloaddition) with the employment of sustainable solvents as reaction media. In most of the presented examples, the use of water, glycerol (*Gly*), or deep eutectic solvents (*DESs*) as non-conventional reaction media allowed not only to recycle the catalytic system (thus reducing the amount of the copper catalyst needed per mole of substrate), but also to achieve higher conversions and selectivities when compared with the reaction promoted in hazardous and volatile organic solvents (*VOCs*). Moreover, the use of the aforementioned green solvents also permits the improvement of the overall sustainability of the Cu-catalyzed 1,3-dipolar cycloaddition process, thus fulfilling several important principles of green chemistry.

## 1. Introduction

The employment of copper catalysts, as opposed to the use of stoichiometric amounts of other typical promoters (like Brønsted acids, bases, radicals, etc.), has permitted the discovery of a plethora of new synthetic protocols within the toolbox of organic chemists [1]. Thus, the introduction of these Cu-catalyzed methodologies in different fields (ranging from asymmetric synthesis [2] to cross-coupling processes [3]) paved the way for the convenient and successful synthesis of high value-added organic products (like for example agrochemical or pharmaceutical commodities). Moreover, Cu is usually considered as “the metal of choice” in organic transformations being catalyzed by coinage metals [4]. This is due to the following properties: (i) it is an earth-crust abundant first row transition metal, and thus cheaper than other second or third row precious metals (like for example, Pd, Pt, Rh, Ir, or Au); (ii) it presents a wide variety of oxidations states (Cu(0), Cu(I), Cu(II) or even Cu(III)) with different coordination environments (for example, linear, square planar or tetrahedral); (iii) it accommodates both hard or soft donor ligands in its coordination sphere, and forms σ- or/and π-interactions with unsaturated organic substrates (like alkenes or alkynes); (iv) it is an important metal in Biology (present in enzymes and proteins) [5], also being the less toxic first row metal (together with Fe); and (v) it is active under both homogenous or heterogeneous [6] conditions.

These advantageous properties make copper catalysts suitable candidates to be used in organic synthesis, thus being not surprising to observe a positive slope in the tendency of the number of publications on this topic since 2010 (see Figure 1).

With no doubt, one of the most prolific and successful applications of copper catalysts in organic synthesis are the so-called CuAAC reactions (Copper-catalyzed azide-alkyne cycloaddition) that belong to the family of “click chemistry” procedures. First coined by Sharpless back in 2001 [7], this concept appeared to design thermodynamically driven chemical processes that allow the direct synthesis of highly complex organic architectures by starting from simple and readily-available organic building blocks. Moreover, to be considered “click”, these reactions should be in accordance with the following principles: (i) tolerance of moisture and oxygen (i.e., standard bench conditions); (ii) “solventless” conditions or using a benign solvent is required; (iii) the isolation of the final products must be straightforward (i.e., nearly quantitative and without tedious steps of purification); and (iv) should occur with total regio- and stereoselectivity. In this sense, the aforementioned CuAAC (independently reported by the groups of Meldal [8] and Sharpless [9]) is considered as one of the most genuine examples of this “click chemistry” concept. This major discovery is based on the previously reported Huisgen cycloaddition, a synthetic tool that proceeds slowly under thermal conditions giving rise to a mixture of the corresponding 1,4- and 1,5-disubstituted 1,2,3-triazoles (see Scheme 1a) [10]. Meldal and Sharpless found that just by simple addition of a copper catalyst, this atom economic 1,3-dipolar cycloaddition leads to the 1,4-triazole as the only product of reaction, thus fulfilling now the principles of “click chemistry” (see Scheme 1b). Although either metal-free [11] or Ag-[12], Ru-[13], Ir-[14] and Ni-catalyzed [15] cycloadditions of azides and alkynes have been reported, only Cu-catalysts have found a wide chemical application [16], like: (i) drug discovery and development; (ii) synthesis of peptides, carbohydrate-based molecules or nucleosides/nucleotides; (iii) polymer science; and (iv) macrocyclic chemistry.

In accordance with the originally coined definition of “click chemistry” and the requirements for a chemical process to be considered “click” [7], the desired transformation should take place under solventless reaction conditions. However, it is now well-established that solvents usually present a concomitant positive effect in the overall synthetic process, the so-called “solvent effect” [17]. This unquestionable positive solvent effect is generally associated with the capability of solvents to: (i) stabilize both metallic catalysts and transient intermediates; (ii) reduce unwanted side reactions; and (iii) help in the control of the heat flow. Thus, in most cases, the best option to work under strict “click” conditions is to replace traditional and hazardous volatile organic solvents (*VOCs*) with an environmentally friendly reaction media. In theory, these solvents to be considered green should be [18]: (i) biorenewable and biodegradable; (ii) cheap and easily available; (iii) non-toxic for both the environment and the human beings; (iv) safe (non-flammable, low vapor pressure); and (v) capable of dissolving a wide range of chemicals. With these strict requirements in mind, it is not surprising to find that the pool of available green and sustainable solvents is quite narrow, being the most typical examples: (i) water [19]; (ii) biomass derived solvents (like glycerol [20], γ-valerolactone [21], 2-methyl-THF [22] or lactic acid [23]); or (*iii*) Deep Eutectic Solvents (*DESs*) [24]. Thus, this mini-review focuses on the recent progress achieved on 1,3-dipolar cycloadditions of azides and alkynes mainly using molecular Cu-catalysts (CuAAC) that take place in glycerol (*Gly*), Deep Eutectic Solvents (*DESs*), or water as sustainable reaction media. 

## 2. Cu-Catalyzed 1,3-Dipolar Cycloaddition of Azides and Alkynes (CuAAC) in Glycerol (*Gly*)

As previously commented, the use of green solvents is mandatory for the development of CuAAC processes under “click” conditions [7]. In this regard, biomass derived solvents are playing a pivotal role in the substitution of traditional volatile and hazardous organic solvents (*VOCs*) by greener, safer and sustainable reaction media [25]. Among all biomass-derived solvents, glycerol (*Gly*, which is obtained as a major by-product from the biodiesel industry [26] or from the conversion of lignocellulose or cellulose into other chemical compounds [27]), has received great attention from the synthetic community as a plausible candidate for the replacement of *VOC* solvents in organic synthesis [20]. Moreover, due to its: (i) inherent physicochemical properties (low toxicity, non-flammability, high polarity, and boiling point); (ii) capability to dissolve both inorganic and organic compounds; (iii) easily separation of organic products and catalysts from the reaction media (that allows catalyst recycling); and (iv) previously reported competence to enhance both selectivity and productivity of given chemical reactions [28], glycerol is considered one of the archetypical prototypes of green and sustainable solvents. In this sense, García-Álvarez and co-workers [29] paved the way to follow in the CuAAC reaction using CuI/glycerol as catalytic system. This unprecedented combination promoted the Huisgen cycloaddition of organic azides with either terminal or internal 1-iodoalkynes at room temperature using low catalyst loadings (1 mol% in Cu), in the presence of air and in the absence of any external base (Scheme 2a). In this work, the authors were able to recycle the aforementioned catalytic system up to six consecutives cycles and avoid the use of *VOC* solvents, as straightforward isolation of the desired triazoles was accomplished by simple filtration of the reaction crude. For comparison, the catalytic activity of CuI in a conventional and hazardous volatile organic solvent (i.e., CH_2_Cl_2_) was also evaluated and longer reaction times (24 h) were needed to achieve quantitative conversions (99%). This example supports clearly our previous affirmation which assessed the importance of green solvents in the design of a synthetic chemical process, thus disclosing a new example of an accelerated organic reaction in environmentally friendly reaction media.

More recently, Schoffstall and co-workers have employed the same catalytic system (CuI/glycerol) for the synthesis of 1,2,3-triazoles containing fluorinated groups (Scheme 2b) [30]. In the same year (2019), Bez et al. proved the beneficial effect of *L*-proline as a ligand to improve the catalytic activity of CuI in CuAAC cycloadditions using glycerol as solvent [31]. This methodology was successfully applied to the synthesis of possible pharmacologically active heterocyclic compounds like, for example, those derived from propargylated dihydroartemisinin (Scheme 2c).

Contemporaneously to and independent of the pioneering work by García-Álvarez et al., Gómez et al. demonstrated that glycerol by itself (without the help of any transition metal catalysts) is able to promote the Huisgen reaction of internal alkynes and organic azides when employing both microwave irradiation and thermal treatment (warming up to 100 °C) [32,33] (Scheme 3a). For the case of unsymmetrically substituted alkynes, the authors observed the formation of an almost equimolar mixture of both regioisomers.

After this seminal work, the same authors reported the possibility to use Cu(I)-based-nanoparticles as highly efficient catalyst for the cycloaddition of organic azides and terminal alkynes [34]. In this work, Gómez, Pericàs and co-workers studied deeply the role of different aliphatic and aromatic amines as co-catalysts for CuAAC reactions finding that an excess of oleylamine (5 mol%) favors the in-situ generation of Cu-nanoparticles when using CuI (1 mol%) as pre-catalyst (see Scheme 3b). Remarkably, the formation of these Cu(I)-nanoparticles was also observed in other highly polar solvents like water or dioxane. More recently, the Gómez group has also reported the three-component version of AAC cycloadditions (by using the mixture NaN_3_/PhCH_2_Br as a source of benzyl azide) in pure glycerol (at 80 °C) by employing small and well-dispersed zerovalent PdCu bimetallic nanoparticles (PdCuNPs, mean diameter, ca. 3−4 nm) stabilized by polyvinylpyrrolidone (PVP) as catalysts (see Scheme 3c) [35].

Apart from glycerol, other biosourced alcohol-type solvents like polyethylene glycol (PEG) have been employed in CuAAC reactions [36,37,38,39,40,41,42,43]. In this sense, PEG-400 has been used as environmentally friendly and inexpensive solvent thanks to its: (i) low toxicity; (ii) non-volatility; (iii) recyclability and biodegradability; (iv) thermal stability; and (v) commercial availability [44]. Thus, the first work in this field was reported back in 2006 by Sreedhar and co-workers who describe an efficient and reliable CuI-catalyzed three-component reaction that employs terminal alkynes, NaN_3_ and Baylis-Hillman acetates as starting materials for the high-yielding synthesis of 1,4-disubtitutes triazoles in PEG at room temperature (see Scheme 4) [36]. At this point it is important to note that better yields of the desired triazoles were obtained when hazardous and volatile organic solvents (like THF, CH_3_CN or DMSO) were replaced by PEG. After this pioneering work, the aforementioned CuI/PEG catalytic system was applied for the synthesis of: (i) β-hydroxy or *N*-tosylamino 1,2,3-triazole scaffolds [37]; (ii) biologically active bis(indolyl)methane derivatized 1,4-disubstituted 1,2,3-bistriazoles [38]; (iii) 1,4-diaryl-1*H*-1,2,3-triazoles by reaction of diaryliodonium salts, NaN_3_ and terminal alkynes [39]; and (iv) 4-substituted-1,2,3-triazole analogues of azole antifungal agents [40]. Not only CuI, but also the archetypical catalytic mixture CuSO_4_·5H_2_O/ascorbic acid has been fruitfully applied in CuAAC processes devoted to the synthesis of 1,2,3-triazole tethered benzimidazo[1,2-*a*]quinolines [41] or 1*H*-1,2,3-triazole tethered pyrazolo[3,4-b]pyridin-6(7*H*)-ones [42] by employing PEG as sustainable solvent at high temperatures (100–120 °C). Finally, Astruc and co-workers nicely described the use of PEG as stabilizer for copper nanoparticles (CuNP) which are catalytically active in CuAAC processes in a mixture water/*^t^*BuOH and in the presence of air [43].

## 3. Cu-Catalyzed 1,3-Dipolar Cycloaddition of Azides and Alkynes (CuAAC) in *Deep Eutectic Solvents* (*DESs*)

Deep Eutectic Solvents (*DESs*) [24] are defined as the result of the combination of two (or even three) chemical substances which are able to form a new eutectic mixture (with a melting point below that of its individual components) through the formation of a tridimensional hydrogen-bond network [45]. One of the most commonly used components in the synthesis of these eutectic mixtures is the biorenewable and biodegradable ammonium salt choline chloride (*ChCl*, vitamin B4) [46]. In combination with different hydrogen bond donors like: (i) naturally occurring polyols (glycerol, ethylene glycol or carbohydrates) [47], (ii) biorenewable organic acids [48], or (iii) urea [49]; choline chloride (*ChCl*) is able to form sustainable and liquid eutectic mixtures which have found application in a plethora of different chemical fields [50,51,52]. As expected, and when the concept of Deep Eutectic Solvents crossed the field of traditional organic synthesis, CuAAC was also assayed in these sustainable reaction media. In this sense, König et al. were the real pioneers in this field back in 2009 by reporting the CuI-catalyzed cycloaddition of phenylacetylene with benzyl azide for the synthesis of the archetypical 1,4-disubstituted 1,2,3-triazoles [53]. However, it should be mentioned that: (i) quite drastic conditions (85 °C, 5 mol% of CuI and 5 h of reaction) were needed to obtain the desired triazole in high yield (93%, see Scheme 5a), and (ii) no recycling studies of the catalytic system in *DESs* were reported. In contrast, employing a catalytic system based on the combination of CuSO_4_ with sodium ascorbate as reducing agent (typically used in other CuAAC reactions), the yield of the desired triazole only reached 84%. The use of the ternary eutectic mixture *D*-sorbitol/urea/NH_4_Cl, as sustainable solvent, also allowed the in-situ synthesis of the benzyl azide (starting from PhCH_2_Br and NaN_3_) in the corresponding three-component version of the CuAAC process. Interestingly, the aforementioned CuAAC process proceeded slightly better when the *L*-carnitine-based eutectic mixture *L*-carnitine/Urea was used as sustainable solvent (see Scheme 5b). This experimental observation was attributed to the formation of a coordination complex between CuI and *L*-carnitine that prevented the unwanted oxidation of Cu(I) into Cu(II). Lately, García-Alvarez and co-workers demonstrated that the use of the eutectic mixture formed by choline chloride and glycerol (1*ChCl*/2*Gly*) allowed the use of milder reaction conditions (room temperature and 1 mol% of CuI) although longer reaction times (14 h) were needed to reach quantitative yields (97%, Scheme 5c) [29]. At this point, it is important to note that, when the eutectic mixture 1*ChCl*/2*Gly* was replaced by volatile and hazardous CH_2_Cl_2_, under the same catalytic conditions, a longer reaction time (24 h) was needed to achieve quantitative conversion.

More recently, and bearing in mind the positive effect observed in CuAAC when: (i) the eutectic mixture 1*ChCl*/2*Gly* was employed as sustainable solvent, and (ii) nitrogenated ligands (amines) are added as co-catalyst to the reaction media; Handy et al. reported the CuI-catalyzed cycloaddition of in-situ generated aryl azides with terminal alkynes by employing the aforementioned 1*ChCl*/2*Gly* eutectic mixture as sustainable solvent in the presence of *N*,*N*-dimethylethylenediamine (DMEAD, 20 mol%) as co-catalyst. Although the catalytic system was recycled up to 4 consecutive cycles, quite drastic reaction conditions (75 °C, 10 mol% of CuI and 5–10 h, see Scheme 6) were still needed to achieve moderate to good yields (40–88%) [54,55].

## 4. Cu-Catalyzed 1,3-Dipolar Cycloaddition of Azides and Alkynes (CuAAC) in Water

In opposite to the more traditional *VOCs*, the use of water as a reaction media offers multiple advantages such as: (i) its non-toxicity and non-flammability; (ii) elevated boiling point; (iii) high availability and low price; and (iv) non-miscibility with organic compounds that enables easy product separation and catalyst recycling upon decantation. Taken together, and in agreement with the principles of green chemistry, water [19] emerges as one of the most attractive solvents in terms of sustainability and often represents the solvent of choice when dealing with CuAAC processes. This is in part due to the pioneering work reported by Sharpless and Fokin in 2002 occurring in aqueous solutions (2:1 mixture of water and *^t^*BuOH) [9] that inspired a plethora of international research groups worldwide to study CuAAC reactions in this environmentally friendly solvent. On the other hand, and keeping in mind that the state of the art for the CuAAC processes has been reviewed recently [56,57,58,59,60,61,62], this section focuses on most representative examples appeared since 2015 merging the use of molecular copper catalysts and pure aqueous media.

In spite of the well documented viability of CuAAC reactions in pure water under ligand- and additive-free conditions [63], the finding of highly efficient CuAAC procedures catalyzed by commercially available Cu salts is an active research area. In this sense, impressive substrate scope was reached recently by Jiang and Xu using Cu(II) catalysts in very low metallic charges (see Scheme 7a,b) [64,65]. Both Cu(II) catalysts (CuSO_4_·5H_2_O [64] or Cu(acac)_2_ [65]) required harsh conditions. In contrast, they proved efficient for the three-component synthesis of 1,4-disubstituted 1,2,3-triazoles starting from terminal alkynes, NaN_3_ and organic halides (R-X).

Milder conditions were required using CuCl_2_ as catalyst in combination with an organic photosensitizer (eosin Y disodium salt (EY)) and green LED irradiation (Scheme 7c) [66]. The CuCl_2_/EY photocatalytic system enabled the isolation of a broad family of 1,4-disubstituted 1,2,3-triazoles in moderate to excellent yields (49–100%) and the aforementioned catalytic system was recycled up to three consecutive runs [66].

An additional example of ligand-free CuAAC protocols in neat water includes the use of ultrasound irradiation [67] to enhance the activity of a Cu(I) salt (i.e., CuCl) as catalyst (Scheme 8). In this sense, Chen, Qu and co-workers used CuCl (10 mol%) to build 1,4-disubstituted 1,2,3-triazoles at room temperature upon ultrasound irradiation (150 W), including a family of triazoles generated from coumarine-derived alkynes or azides thereby granting access to highly functionalized heterocycles that were isolated in ca. 90% yield (see Scheme 8a,b) [67]. Remarkably, lower yields were reached in the absence of ultrasonic power, even in the presence of trimethylamine as co-catalyst, or when using mixtures H_2_O/*VOCs* or pure organic solvents (DMSO, DMF, toluene or *^t^*BuOH) instead of neat water as a reaction medium.

The efficiency of the CuAAC reactions in aqueous media can be improved using heteroatom-based donor ligands that enhance the solubility and stability of the Cu catalyst. Classical examples of this strategy combined CuX species (X = I, Br) and NEt_3_ [68], *N*-alkylimidazoles [69] or thioethers [70,71]. In 2014, Sarma and colleagues carried out the cycloaddition of organic azides with terminal alkynes in excellent yields (82–92%) catalyzed by CuI (0.5 mol%) and the hydroquinidine-1,4-phthalazinediyl ligand ((DHQD)_2_PHAL; 1 mol%) in pure water (see Scheme 9a) [72]. The efficiency of the CuI/(DHQD)_2_PHAL catalytic system in the 1,3-dipolar cycloaddition of phenylacetylene and benzyl azide was completely inhibited in absence of ligand, and the beneficial effect of water was demonstrated by the lower activity of the aforementioned catalytic system when using volatile organic solvents such as THF, CH_2_Cl_2_, DMSO, DMF, acetone, toluene, or mixtures H_2_O/*^t^*BuOH [72]. Cu(OAc)_2_ and hydrazine monohydrate (NH_2_NH_2_·H_2_O) was used by Xu and co-workers to prepare a family of 1,4-disubstituted 1,2,3-triazoles in neat water (see Scheme 9b) [73]. In this case, the hydrazine behaves as stoichiometric reducing agent to generate Cu(OAc) species in-situ that underwent spontaneous decomposition in aqueous media to produce spherical Cu_2_O-NPs with size distribution ranging from 400 to 500 nm (characterized by powder XRD, SEM and TEM). It should be noted that the excellent catalytic activity and robustness of discrete Cu_2_O species in AAC reactions occurring in neat water at room temperature and in absence of additives was noticed back in 2011 by Bi and Zhang [63], also pointing to the key role of water since only traces (DMSO, THF, and MeOH) or very low conversions (< 20%; CHCl_3_) were reached using volatile organic solvents. Interestingly, the Cu(OAc)_2_ to Cu_2_O-NPs conversion co-produced HOAc as by-product, that suggested the beneficial role of acidic media for the CuAAC reaction to proceed [73]. Built on these observations, the same group developed the fast and high-yielding 1,3-dipolar cycloaddition of organic azides with terminal alkynes catalyzed by Cu_2_O (0.5 mol%) in neat water using carboxymethylpullulans (CMP; 5 mol%) as a mild source of protons (see Scheme 9c) [74]. Remarkably, the Cu_2_O/CMP catalytic system was efficiently recycled up to 6 consecutives runs, and might be recycled six additional runs upon acidic treatment that reactivates the CMP co-catalyst. Joint work performed by the groups of Pericàs and Gómez proved the pivotal role of long chain alkyl amines in CuAAC reactions taking place in polar solvents such as alcohols, dioxane and pure water [34]. For the case of water, the authors clearly illustrated the need of HN(octyl)_2_ or oleylamine in catalytic amounts to reach nearly quantitative yield of 1-benzyl-4-phenyl-1*H*-1,2,3-triazole, whereas negligible amounts of the cycloaddition product where isolated in presence of NEt_3_ (5 mol%) or in the absence of any additive (13%; see Scheme 9d) [34].

Copper acetylides are commonly invoked as catalytic intermediates in CuAAC reactions, which makes the synthesis of functionalized 1,2,3-triazoles from internal alkynes typically unreachable. This limitation is frequently circumvented using internal iodoalkynes [75], thus allowing for further modification of the triazole skeleton. An elegant approach to 5-iodo-1,2,3-triazoles taking place in water via three-component reaction between terminal alkynes, organic azides and NaI was discovered recently by Li, Cui and Zhang (see Scheme 10) [76]. After careful optimization of the reaction conditions (10 mol% of CuI along with equimolar amounts of NaI, DIPEA (diisopropylethylamine), TBACl (tetrabutylammonium chloride), Selectfluor (1.2 equiv of each), and mild heating), the authors isolated a number of 5-iodo-1,2,3-triazoles in yields ranging from 72% to 91%. By applying this catalytic cocktail, broad substrate scope and functional group tolerance was reached, including the incorporation of relevant motifs such as sugars, fluorophore groups, or peptides.

The use of well-defined Cu complexes as catalysts for the CuAACs in aqueous media has been extensively studied using phosphorus-donor ligands [77,78], *N*-Heterocyclic Carbenes (NHC) [79,80,81], or PTA-derived iminophosphoranes (PTA = 1,3,5-Triaza-7-phosphaadamantane) [82,83]. To the best of our knowledge, the first isolated and fully characterized copper species bearing an *N*-donor ligand that proved efficient in CuAAC reactions using pure water as solvent was reported by Pericàs back in 2009 using a 1:1 complex of CuCl and the tris(1-benzyl-1*H*-1,2,3-triazol-4-yl)methanol ligand [84]. Pursuing this approach, Pericàs and co-workers prepared a family of Cu-complexes (**1a**–**f**) bearing tris(triazolyl)methane ligands that catalyzed the cycloaddition of benzyl azide with phenylacetylene at room temperature in neat water as a solvent (see Scheme 11a) [85]. A comparative study using the simplest and hydrophilic Cu species **1b** (R = H; R’ = Ph), which behaved as the most efficient catalyst for the model reaction between phenylacetylene and benzyl azide, has revealed the positive impact of water (vs. hexanes, toluene, CH_2_Cl_2_, THF or CH_3_CN) on the catalytic activity of **1b [85]**. The (2-pyrrolecarbaldiminato)-Cu(II) catalyst **2** allowed the three-component and high-yielding synthesis of 1,4-disubstituted 1,2,3-triazoles (see Scheme 11b) [86]. Interestingly, the scope is restricted to the use of benzyl halides which leaves the inert aryl-Cl and/or aryl-Br bonds intact for further functionalization [86]. The groups of Mahmudov and Pombeiro carried out the three-component coupling of phenylacetylene, benzyl bromide and NaN_3_ in neat water catalyzed by the Cu(II)-catalyst **3**^.^2H_2_O (3 mol%) upon microwave irradiation (10 W) and strong heating (125 °C; see Scheme 11c) [87]. Nevertheless, the harsh conditions only permitted the isolation of the model 1-benzyl-4-phenyl-1*H*-1,2,3-triazole in 74% yield. A last example of AAC enabled by well-defined [(NHC)CuCl] species **4** was given by the group of Szadkowska [88]. In order to enhance the solubility of **4** in aqueous media, the authors used a hydrophilic NHC ligand containing the theophylline backbone and an ammonium moiety. This ligand design strategy proved right and was key to enhance the catalytic activity of **4** in neat water that surpassed the one found in non-polar organic solvents (toluene or ethers), alcohols (glycerol or MeOH), CH_2_Cl_2_, DMSO or mixtures H_2_O/*^t^*BuOH. In addition, the three-component synthesis of functionalized triazoles was accomplished using low copper loadings (1 mol% of **4**) and mild conditions (room temperature; see Scheme 11d) [88].

In 2014, Astruc and colleagues performed the 1,3-dipolar cycloaddition of terminal alkynes and organic azides enabled by either [Cu(hexabenzyltren)Br] (tren = triaminoethylamine; complex **5** in Scheme 12) or the Sharpless-Fokin catalyst [CuSO_4_·5H_2_O and sodium ascorbate (NaAsc) in Scheme 12] using micellar catalysis at nearly room temperature [89]. To meet success, the amphiphilic dendrimer **6** that bears 27 triethylene glycol (TEG) units and nine intradentritic triazoles was used in catalytic amounts allowing the encapsulation of the Cu(I) active sites (Scheme 12) [89]. The resulting Cu(I)-containing nanoreactor permitted the isolation of several 1,4-disubstituted 1,2,3-triazoles in excellent yields and with excellent TON values (up to 510000). Moreover, at this point it is important to note that the authors reported the recycling (up to 10 consecutive runs) of the amphiphilic dendrimer **6** using the Cu(I)-catalyst **5**.

A year later, a similar approach was employed by Shin and Lim to isolate a family of glycidyl triazolyl polymers via CuAACs using CuSO_4_·5H_2_O (5 mol%), sodium ascorbate (NaAsc, 15 mol%) and *β*-cyclodextrin (*β-CD*; 2.5 mol%) using neat water as solvent (see Scheme 13) [90]. The replacement of *β-CD* by the zwitterionic surfactant Betaine permitted the groups of Lee and Lim to achieve the quantitative cycloaddition of phenylacetylene and benzyl azide combining CuSO_4_·5H_2_O (25 ppb), sodium ascorbate (15 mol%) and Betaine (5 mol%; see Scheme 14) [91]. Upon these conditions, very impressive TON and TOF values (32,000,000 and 1,333,333 h^−1^, respectively) were accomplished at nearly room temperature, and the micellar aqueous medium was reused up to three cycles [although higher copper charges (200 ppm) and longer reaction times were required to reach completion]. In addition, the catalytic system was applied to the isolation of 1-benzyl-4-phenyl-1*H*-1,2,3-triazole in a gram scale using 2.5 ppm of Cu-catalyst (95% yield), and the scope of the reaction was extended to a family of aryl, alkyl, and ferrocenyl acetylenes, furnishing the corresponding triazoles in 76–98% yield upon very mild conditions (copper loadings of 25–200 ppm and heating at 30 °C).

Last but not least, the group of Scarso proved that micellar catalysis can be applied to the CuAAC reaction in its three-component version. Thus, the selective coupling of terminal alkynes, organic bromides and NaN_3_ efficiently proceeds using the lipophilic catalyst [Cu(IMes)Cl] (1 mol% Cu; IMes = 1,3-bis(2,4,6-trimethylphenyl)imidazol-2-ylidene)) and the commercially available surfactants named sodium lauryl sulfosuccinate (SLS) or *DL*-α-tocopherol methoxypolyethylene glycol succinate (TPGS-750-M) giving access to the corresponding triazoles in variable yields (51–98%; see Scheme 15) [92]. It is important to note that the efficiency of [Cu(IMes)Cl] upon micellar conditions to achieve the 1,3-dipolar cycloaddition between 1-octyne and benzyl azide notably outperforms its catalytic activity using volatile organic solvents such as CH_2_Cl_2_ or MeOH, that only produced the desired triazole in ca. 5%. Unfortunately, attempts to make possible the recycling of the micellar catalytic system proved unfruitful and the model triazole was isolated in only 27% yield.

## 5. Conclusions

“Click” criteria has been designed in order to reduce chemical waste production and favor the easy manipulation and isolation of the targeted high-value commodities. Amongst all metal-catalyzed transformations coming into existence upon strict “click” criteria, the copper-catalyzed azide-alkyne cycloaddition (CuAAC) occupies a preferential place, and represents one of the most reliable synthetic tools in organic chemistry, leading to functionalized triazoles in a one-pot manner. This mini review collects remarkable contributions to the CuAAC reaction obeying the “click” criteria and occurring in environmentally friendly solvents namely: glycerol (*Gly*), Polyethylene glycol (PEG), Deep Eutectic mixtures *(DESs)* or water. As a general trend, CuAAC reactions can be achieved using either commercially available Cu-salts or in-house made Cu complexes containing an ancillary ligand, usually under mild reaction conditions. Although less common, different ways to enhance the catalytic activity of simple CuX and CuX_2_ salts in AACs include: (i) the combination of an organic photosensitizer and green LED irradiation; (ii) the controlled protonation of the Cu_2_O-catalyst using mild proton sources; (iii) the design of micellar aqueous solutions; and (iv) the use of sonication power or MW irradiation. By using these approaches, more efficient procedures were developed allowing to: (i) decrease the catalyst loadings up to ppb levels providing excellent TON and TOF values (32,000,000 and 1,333,333 h^−1^, respectively); (ii) improve the substrate scope; and (iii) achieve the recovery of the Cu-catalyst and perform recycling experiments. In spite of growing interest in the field of CuAAC reactions using novel environmentally-friendly solvents, considerable room is still available for the improvement of this technology using the emerging “green solvents” named glycerol (Gly) and deep eutectic mixtures (DESs) as only a handful of examples have been reported in the literature.

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
