# Peer review of "Recent Progress of Cu-Catalyzed Azide-Alkyne Cycloaddition Reactions (CuAAC) in Sustainable Solvents: Glycerol, Deep Eutectic Solvents, and Aqueous Media"

_molecules, 2020, doi:10.3390/molecules25092015_

Round 1
Reviewer 1 Report
The present review is quite important, especially for the researchers working in the field of synthetic organic chemistry. The topic is actual as the search for "Green chemistry" conditions, especially, environment-friendly solvents, is highly demanded. The material of the review is given logically, however, there are some points which should be considered.
1) It is well known that poly(ethylene) glycols are widely used for many reactions, including catalytic, and are in the trend of ecologically relevant solvents. However, no mention of them has been done in this review. I am not sure that such material is abundant, however, I have found easily some information: doi 10.1039/c7cc02504a, doi 10.5772/intechopen.83776. Authors should clarify the state of the art concerning PEGs as solvents.
2) Of course, additional information on the subject and related themes is welcome, but it seems that too many references are given in the general introduction and in shorter introductions to each part which do not deal directly with the main topic of the review. Either their number can be diminished or at least the text can be reduced. Otherwise it becomes tiresome. Note that less than a half of the references (41 of 91) are essential for the review's main theme.
3) Conclusions are enormous! They must be as short and precise as possible (one small paragraph) and by no means contain references which have been already cited.
4) No comparison with convenient solvents (like DMF) was provided by the authors. Of course, it is impossible to compare all cited reactions but in the case of each "green" solvent some examples could by of great interest as the authors noted in the beginning that the role of the solvents could not be overestimated.
5) The manuscript should be checked for possible grammatical errors (like "needed to achieved quantitative conversions (lines 195-196)") and style can also be improved ("the synthesis of 1,2,3-triazoles displaying fluorinated arenes as substituents" (lines 126-127); "that guided many actors to pursue the CuAAC reactions" (lines 208-209), especially the bureaucratic-style sentences like "The resulting Cu(I)-containing nanoreactor allowed for: i) the isolation of several 1,4-disubstituted 1,2,3-triazoles in excellent yields; ii) the obtaining of excellent TON values (up to 510000); and iii) the recycling of the amphiphilic dendrimer 6 using the Cu(I)-catalyst 5 up to 10 consecutive runs." (lines 315-317)).
Author Response
The present review is quite important, especially for the researchers working in the field of synthetic organic chemistry. The topic is actual as the search for "Green chemistry" conditions, especially, environment-friendly solvents, is highly demanded. The material of the review is given logically, however, there are some points which should be considered.
Answer: We are grateful to this referee for highlighting the importance of our work and also for mention that the material is given logically.
Question 1: It is well known that poly(ethylene) glycols are widely used for many reactions, including catalytic, and are in the trend of ecologically relevant solvents. However, no mention of them has been done in this review. I am not sure that such material is abundant, however, I have found easily some information: doi 10.1039/c7cc02504a, doi 10.5772/intechopen.83776. Authors should clarify the state of the art concerning PEGs as solvents.
Answer: We thank this reviewer for pointing this out and we totally agree with her/his comment. Thus, we have introduced in our mini-review a new paragraph (lines 144-164) that resumes the state-of the-art of CuAAC reaction in PEG. This paragraph describes the articles proposed by the reviewer and 7 more works. Moreover, a new scheme (Scheme 4) has been included in the main text of our manuscript (view infra).
Page 5, lines 144-164:
Apart from glycerol, other biosourced alcohol-type solvents like polyethylene glycol (PEG) have been employed in CuAAC reactions [36-43]. In this sense, PEG-400 has been used as environmentally-friendly and inexpensive solvent thanks to its: i) low toxicity; ii) non-volatility; iii) recyclability and biodegradability; iv) thermal stability; and v) commercial availability [44]. Thus, the first work in this field was reported back in 2006 by Sreedhar and co-workers who describe an efficient and reliable CuI-catalyzed three-component reaction that employs terminal alkynes, NaN3 and Baylis-Hillman acetates as starting materials for the high-yielding synthesis of 1,4-disubtitutes triazoles in PEG at room temperature (see Scheme 4) [36]. At this point it is important to note that better yields of the desired triazoles were obtained when hazardous and volatile organic solvents (like THF, CH3CN or DMSO) were replaced by PEG. After this pioneering work, the aforementioned CuI/PEG catalytic system was applied for the synthesis of: i) β-hydroxy or N-tosylamino 1,2,3-triazole scaffolds [37]; ii) biologically active bis(indolyl)methane derivatized 1,4-disubstituted 1,2,3-bistriazoles [38]; iii) 1,4-diaryl-1H-1,2,3-triazoles by reaction of diaryliodonium salts, NaN3 and terminal alkynes [39]; and iv) 4-substituted-1,2,3-triazole analogues of azole antifungal agents [40]. Not only CuI but also the archetypical catalytic mixture CuSO4·5H2O/Ascorbic Acid has been fruitfully applied in CuAAC processes devoted to the synthesis of 1,2,3-triazole tethered benzimidazo[1,2-a]quinolines [41] or 1H-1,2,3-triazole tethered pyrazolo[3,4-b]pyridin-6(7H)-ones [42] by employing PEG as sustainable solvent at high temperatures (100-120 ºC). Finally, Astruc and co-workers nicely described the use of PEG as stabilizer for copper nanoparticles (CuNP) which are catalytically active in CuAAC processes in a mixture water/tBuOH and in the presence of air [43].
Scheme 4 CuI-catalyzed synthesis of 1,4-disubstituted 1,2,3-triazoles starting from Baylis–Hillman acetates, terminal alkynes and sodium azide using PEG as sustainable solvent.
- Sreedhar, B., Reddy, P. S., Kumar, N. S. Cu(I)-catalyzed one-pot synthesis of 1,4-disubstituted 1,2,3-triazoles via nucleophilic displacement and 1,3-dipolar cycloaddition. Tetrahedron Lett. 2006, 47, 3055-3058. DOI: 1016/j.tetlet.2006.03.007.
- Kumaraswamy, G., Ankamma, K., Pichaiah, A. Tandem Epoxide or Aziridine Ring Opening by Azide/Copper Catalyzed [3+2] Cycloaddition: Efficient Synthesis of 1,2,3-Triazolo β-Hydroxy or β-Tosylamino Functionality Motif. Org. Chem. 2007, 72, 9822-9825. DOI: 10.1021/jo701724f.
- Damodiran, M., Muralidharan, D., Perumal, P. T. Regioselective synthesis and biological evaluation of bis(indolyl)methane derivatized 1,4-disubstituted 1,2,3-bistriazoles as anti-infective agents. Med. Chem. Lett. 2009, 19, 3611-3614. DOI: 10.1016/j.bmcl.2009.04.131.
- Kumar, D. P., Reddy V. B. An Efficient, One-Pot, Regioselective Synthesis of 1,4-Diaryl-1H-1,2,3-triazoles Using Click Chemistry. Synthesis, 2010, 1687-1691. DOI: 10.1055/s-0029-1218765.
- Pericherla, K., Khedar, P., Khungar, B., Kumar, A. Click chemistry inspired structural modification of azole antifungal agents to synthesize novel ‘drug like’ molecules. Tetrahedron Lett. 2012, 53, 6761-6764. DOI: 1016/j.tetlet.2012.09.129.
- Nagesh, H. N., Suresh, A., Reddy, M. N., Suresh, N., Subbalakshmia, J., Sekhar, K. V. G. C. Multicomponent cascade reaction: Dual role of copper in thesynthesis of 1,2,3-triazole tethered benzimidazo[1,2-a]quinoline and their photophysical studies. RSC Adv. 2016, 6, 15884-15894. DOI: 1039/C5RA24048D.
- Sindhu, J., Singh, H., Khurana, J. M., Bhardwaj, J. K., Saraf, P., Sharma, C. Synthesis and biological evaluation of some functionalized 1H-1,2,3-triazole tethered pyrazolo[3,4-b]pyridin-6(7H)-ones as antimicrobial and apoptosis inducing agents. Chem. Res. 2016, 25, 1813-1830. DOI: 10.1007/s00044-016-1604-0.
- Fu,, Martínez, A., Wang, C., Ciganda, R., Yate, L., Escobar, A., Moya, S., Fouquet, E., Ruiz, J., Astruc, D. Exposure to air boosts CuAAC reactions catalyzed by PEG-stabilized Cu nanoparticles. Chem. Commun. 2017, 53, 5384-5387. DOI: 10.1039/c7cc02504a.
- Chen, J., Spear, S. K., Huddleston, J. G., Rogers, R. D. Polyethylene glycol and solutions of polyethylene glycol as green reaction media. Green Chem. 2005, 7, 64-82. DOI: 1039/B413546F.
Question 2: Of course, additional information on the subject and related themes is welcome, but it seems that too many references are given in the general introduction and in shorter introductions to each part which do not deal directly with the main topic of the review. Either their number can be diminished or at least the text can be reduced. Otherwise it becomes tiresome. Note that less than a half of the references (41 of 91) are essential for the review's main theme.
Answer: We are grateful to this reviewer for this comment that will improve our manuscript. Thus, and following her/his recommendations, we have diminished the number of references in the general introduction and also in the introduction of each part of the mini-review. In the same line, we have also shortened the general introduction.
Question 3: Conclusions are enormous! They must be as short and precise as possible (one small paragraph) and by no means contain references which have been already cited.
Answer: We thank this reviewer for pointing this out. Thus, and following her/his advice we have both shortened the conclusions and eliminated all the references.
Page 14: “Click” criteria has been designed in order to reduce chemical waste production and favor the easy manipulation and isolation of the targeted high-value commodities. Amongst all metal-catalyzed transformations coming into existence upon strict “Click” criteria, the Copper-catalyzed Azide-Alkyne Cycloaddition (CuAAC) occupies a preferential place, and represents one of the most reliable synthetic tools in organic chemistry, leading to functionalized triazoles in a one-pot manner. This mini-review collects remarkable contributions to the CuAAC reaction obeying the “Click” criteria and occurring in environmentally friendly solvents namely: glycerol (Gly), Polyethylene glycol (PEG), Deep Eutectic Mixtures (DESs) or water. As a general trend, CuAAC reactions can be achieved using either commercially available Cu-salts or in-house made Cu complexes containing an ancillary ligand, usually under mild reaction conditions. Although less common, different ways to enhance the catalytic activity of simple CuX and CuX2 salts in AACs include: i) the combination of an organic photosensitizer and green LED irradiation; ii) the controlled protonation of the Cu2O-catalyst using mild proton sources; iii) the design of micellar aqueous solutions; and iv) the use of sonication power and MW irradiation. By using these approaches, more efficient procedures were developed allowing to: i) decrease the catalyst loadings up to ppb levels providing excellent TON and TOF values (32000000 and 1333333 h-1, respectively); ii) improve the substrate scope; and iii) achieve the recovery of the Cu-catalyst and perform recycling experiments. In spite of the growing interest in the field of CuAAC reactions using novel environmentally-friendly solvents, considerable room is still available for improvement of this technology using the emerging “Green Solvents” named glycerol (Gly) and Deep Eutectic Mixtures (DESs) as only a handful of examples have been reported in the literature.
Question 4: No comparison with convenient solvents (like DMF) was provided by the authors. Of course, it is impossible to compare all cited reactions but in the case of each "green" solvent some examples could by of great interest as the authors noted in the beginning that the role of the solvents could not be overestimated.
Answer: We thank again this reviewer for pointing this out and we totally agree with this comment. Thus, and following her/his advice, we have introduced different comparisons with conventional VOC solvents through all the review:
- page 3 lines 111-115: For comparison, the catalytic activity of CuI in a conventional and hazardous volatile organic solvent (i.e., CH2Cl2) was also evaluated and longer reaction times (24 h) were needed to achieve quantitative conversions (99%). This example supports clearly our previous affirmation which assessed the importance of green solvents in the design of a synthetic chemical process, thus disclosing a new example of an accelerated organic reaction in environmentally-friendly reaction media.
- page 5 lines 151-153: At this point it is important to note that better yields of the desired triazoles were obtained when hazardous and volatile organic solvents (like THF, CH3CN or DMSO) were replaced by PEG
- page 6 lines 197-199: At this point, it is important to note that when the eutectic mixture 1ChCl/2Gly was replaced by volatile and hazardous CH2Cl2, under the same catalytic conditions, longer reaction time (24 h) was needed to achieve quantitative conversion.
- page 8 lines 248-249: when using mixtures H2O/VOCs or pure organic solvents (DMSO, DMF, toluene or tBuOH) instead of neat water as a reaction media.
- page 8 lines 258-262: The efficiency of the CuI/(DHQD)2PHAL catalytic system in the 1,3-dipolar cycloaddition of phenylacetylene and benzyl azide was completely inhibited in absence of ligand, and the beneficial effect of water was demonstrated by the lower activity of the aforementioned catalytic system when using volatile organic solvents such as THF, CH2Cl2, DMSO, DMF, acetone, toluene or mixtures H2O/tBuOH [72]
- page 8 lines 267-271: It should be noted that the excellent catalytic activity and robustness of discrete Cu2O species in AAC reactions occurring in neat water at room temperature and in absence of additives was noticed back in 2011 by Xi and Zhang [63], also pointing to the key role of water since only traces (DMSO, THF and MeOH) or very low conversions (<20%; CHCl3) were reached using volatile organic solvents.
- page 10 lines 312-315: A comparative study using the simplest and hydrophilic Cu species 1b (R = R’ = H), which behaved as the most efficient catalyst for the model reaction between phenylacetylene and benzyl azide, has revealed the positive impact of water (vs hexanes, toluene, CH2Cl2, THF or CH3CN) on the catalytic activity of 1b [85].
- page 10 lines 325-328: This ligand design strategy proved right and was key to enhance the catalytic activity of 4 in neat water that surpassed the one found in non-polar organic solvents (toluene or ethers), alcohols (glycerol or MeOH), CH2Cl2, DMSO or mixtures H2O/tBuOH.
- page 13 lines 370-373: It is important to note that the efficiency of [Cu(IMes)Cl] upon micellar conditions to achieve the 1,3-dipolar cycloaddition between 1-octyne and benzyl azide notably outperforms its catalytic activity using volatile organic solvents such as CH2Cl2 or MeOH, that only produced the desired triazole in ca. 5%.
Question 5: The manuscript should be checked for possible grammatical errors (like "needed to achieved quantitative conversions (lines 195-196)") and style can also be improved ("the synthesis of 1,2,3-triazoles displaying fluorinated arenes as substituents" (lines 126-127); "that guided many actors to pursue the CuAAC reactions" (lines 208-209), especially the bureaucratic-style sentences like "The resulting Cu(I)-containing nanoreactor allowed for: i) the isolation of several 1,4-disubstituted 1,2,3-triazoles in excellent yields; ii) the obtaining of excellent TON values (up to 510000); and iii) the recycling of the amphiphilic dendrimer 6 using the Cu(I)-catalyst 5 up to 10 consecutive runs." (lines 315-317)).
Answer: We thank this reviewer for these comments that will improve the grammatical quality of our work. Thus, we have done all the suggested revisions:
- Line 209: we have changed “needed to achieved quantitative conversions” by “needed to achieve moderate to good yields (40-88%)”
- Line 117: we have changed “the synthesis of 1,2,3-triazoles displaying fluorinated arenes as substituents” by “containing fluorinated groups”.
- Lines 221-223: we have changed “that guided many actors to pursue the CuAAC reactions” by “that inspired a plethora of international research groups worldwide to study CuAAC reactions in this environmentally friendly solvent”
- Lines 338-342: we have changed: “The resulting Cu(I)-containing nanoreactor allowed for: i) the isolation of several 1,4-disubstituted 1,2,3-triazoles in excellent yields; ii) the obtaining of excellent TON values (up to 510000); and iii) the recycling of the amphiphilic dendrimer 6 using the Cu(I)-catalyst 5 up to 10 consecutive runs” by “The resulting Cu(I)-containing nanoreactor permitted the isolation of several 1,4-disubstituted 1,2,3-triazoles in excellent yields and with excellent TON values (up to 510000). Moreover, at this point it is important to note that the authors reported the recycling (up to 10 consecutive runs) of the amphiphilic dendrimer 6 using the Cu(I)-catalyst 5.”

Reviewer 2 Report
This is yet another review article on the click chemistry story. The authors should describe well which reviews already have been published, more than now. The first 2.5 pages have been described elsewhere, can be shortened. the Engligh needs correction, I have some detailed remarks
line 22 dipolar
line 34 high value-added
line 48 "has been described its use" reformulate
line 57/8 are the/that belong
Scheme 1 azide has linear N-N-N
line 130 pharmacologically
lines 147 and 178 benzyl azide
Scheme 3(a) there should also be a discussion on the regiochemistry in the text
line 196 40-88 % is not quantitative
There should be more info on the reuse of deep eutectic solvents in this review
page 7-8 magnetized water : this is controversial, I expect a critical discussion and an explanation of the effect
p11 TEG should be also explained in the scheme to make clear that it is connected via oxygen
Why have other biosourced alcohols not been discussed
Author Response
- This is yet another review article on the click chemistry story. The authors should describe well which reviews already have been published, more than now.
Answer: We thank this reviewer for pointing this out. Thus, and following her/his advice we have described the previous reviews already published. See references 56-62:
- Moses, J. E., Moorhouse, A. D. The growing applications of click chemistry. Soc. Rev. 2007, 36, 1249-1262. DOI: 10.1039/B613014N.
- Kappe, C. O., Van der Eycken, E. Click chemistry under non-classical reaction conditions. Soc. Rev. 2010, 39, 1280-1290. DOI: 10.1039/B901973C.
- Thirumurugan, P., Matosiuk, D., Jozwiak, K. Click Chemistry for Drug Development and Diverse Chemical–Biology Applications. Rev. 2013, 113, 4905-4979. DOI: 10.1021/cr200409f.
- Hein, J. E., Fokin, V. V. Copper-catalyzed azide–alkyne cycloaddition (CuAAC) and beyond: new reactivity of copper(i) acetylides. Soc. Rev. 2010, 39, 1302-1315. DOI: 10.1039/B904091A.
- Alonso, F., Moglie, Y., Radivoy, G. Copper Nanoparticles in Click Chemistry. Chem. Res. 2015, 48, 2516-2528. DOI: 10.1021/acs.accounts.5b00293.
- Haldón, E., Nicasio, M. C., Pérez, P. J. Copper-Catalysed Azide–Alkyne Cycloadditions (CuAAC): An Update. Biomol. Chem. 2015, 13, 9528-9550. DOI: 10.1039/c5ob01457c.
- Hassan, S., Müller, T. J. J. Multicomponent Syntheses based upon Copper‐Catalyzed Alkyne‐Azide Cycloaddition. Synth. Catal. 2015, 357, 617-666. DOI: 10.1002/adsc.201400904
- The first 2.5 pages have been described elsewhere, can be shortened. the Engligh needs correction
Answer: We thank again this reviewer for pointing this out and we totally agree with this comment. Following his/her suggestions and those previously pointed out by reviewer 1 in question 2, we have reduced the general introduction. Moreover, and in agreement with the comments of reviewer 1 in question 5, we have done corrections in the English employed in the manuscript.
We have also corrected/commented the following points proposed by the reviewer 2. We are really thankful to this reviewer because thanks to his/her comments our manuscript has been improved:
- line 20 dipolar. Done
- line 31 high value-added. Done
- line 42 "has been described its use" reformulate. Done. We have changed it by “is active under both”
- line 49-50 are the/that belong. Done. We have changed it by “are the so-called CuAAC reactions (Copper-catalyzed Azide-Alkyne Cycloaddition) that belong to the family of “Click Chemistry” procedures”
- Scheme 1 azide has linear N-N-N. Done.
- line 120 pharmacologically. Done.
- lines 141 and 181 benzyl azide. Done.
- Scheme 3(a) there should also be a discussion on the regiochemistry in the text. Done. We have introduced the following sentence (lines 128-129): “For the case of unsymmetrically substituted alkynes, the authors observed the formation of an almost equimolar mixture of both regioisomers”
- line 196 40-88 % is not quantitative. We have changed by (line 209): “needed to achieve moderate to good yields (40-88%) [54,55].”
- There should be more info on the reuse of deep eutectic solvents in this review. We totally agree with this comment. We have commented the reuse of DESs in:
- lines 184-185: “no recycling studies of the catalytic system in DESs were reported”.
- lines 207-208: “Although the catalytic system was recycled up to 4 consecutive cycles”
- page 7-8 magnetized water: this is controversial, I expect a critical discussion and an explanation of the effect. We totally agree with this comment and we also think that the effect of magnetized water is controversial. Thus, and as the authors of the work do not explain with detail this effect, we have decided to eliminate this reference from our mini-review.
- p11 TEG should be also explained in the scheme to make clear that it is connected via oxygen. Done. We have introduced in the scheme the formula of TEG to make clear that is connected to the aromatic ring through the oxygen atom.
- Why have other biosourced alcohols not been discussed. We thank the reviewer for this comment. Following his/her suggestion and also the previous point denoted by reviewer 1 in question 1, we have introduced in our mini-review a new paragraph (lines 144-164) that describes the use of other biosourced alcohols (like PEG) in CuAAC. Moreover, a new scheme (Scheme 4) and 8 new references have been included in the main text of our manuscript (view infra).
Page 5, lines 144-164:
Apart from glycerol, other biosourced alcohol-type solvents like polyethylene glycol (PEG) have been employed in CuAAC reactions [36-43]. In this sense, PEG-400 has been used as environmentally-friendly and inexpensive solvent thanks to its: i) low toxicity; ii) non-volatility; iii) recyclability and biodegradability; iv) thermal stability; and v) commercial availability [44]. Thus, the first work in this field was reported back in 2006 by Sreedhar and co-workers who describe an efficient and reliable CuI-catalyzed three-component reaction that employs terminal alkynes, NaN3 and Baylis-Hillman acetates as starting materials for the high-yielding synthesis of 1,4-disubtitutes triazoles in PEG at room temperature (see Scheme 4) [36]. At this point it is important to note that better yields of the desired triazoles were obtained when hazardous and volatile organic solvents (like THF, CH3CN or DMSO) were replaced by PEG. After this pioneering work, the aforementioned CuI/PEG catalytic system was applied for the synthesis of: i) β-hydroxy or N-tosylamino 1,2,3-triazole scaffolds [37]; ii) biologically active bis(indolyl)methane derivatized 1,4-disubstituted 1,2,3-bistriazoles [38]; iii) 1,4-diaryl-1H-1,2,3-triazoles by reaction of diaryliodonium salts, NaN3 and terminal alkynes [39]; and iv) 4-substituted-1,2,3-triazole analogues of azole antifungal agents [40]. Not only CuI but also the archetypical catalytic mixture CuSO4·5H2O/Ascorbic Acid has been fruitfully applied in CuAAC processes devoted to the synthesis of 1,2,3-triazole tethered benzimidazo[1,2-a]quinolines [41] or 1H-1,2,3-triazole tethered pyrazolo[3,4-b]pyridin-6(7H)-ones [42] by employing PEG as sustainable solvent at high temperatures (100-120 ºC). Finally, Astruc and co-workers nicely described the use of PEG as stabilizer for copper nanoparticles (CuNP) which are catalytically active in CuAAC processes in a mixture water/tBuOH and in the presence of air [43].
Scheme 4 CuI-catalyzed synthesis of 1,4-disubstituted 1,2,3-triazoles starting from Baylis–Hillman acetates, terminal alkynes and sodium azide using PEG as sustainable solvent.
- Sreedhar, B., Reddy, P. S., Kumar, N. S. Cu(I)-catalyzed one-pot synthesis of 1,4-disubstituted 1,2,3-triazoles via nucleophilic displacement and 1,3-dipolar cycloaddition. Tetrahedron Lett. 2006, 47, 3055-3058. DOI: 1016/j.tetlet.2006.03.007.
- Kumaraswamy, G., Ankamma, K., Pichaiah, A. Tandem Epoxide or Aziridine Ring Opening by Azide/Copper Catalyzed [3+2] Cycloaddition: Efficient Synthesis of 1,2,3-Triazolo β-Hydroxy or β-Tosylamino Functionality Motif. Org. Chem. 2007, 72, 9822-9825. DOI: 10.1021/jo701724f.
- Damodiran, M., Muralidharan, D., Perumal, P. T. Regioselective synthesis and biological evaluation of bis(indolyl)methane derivatized 1,4-disubstituted 1,2,3-bistriazoles as anti-infective agents. Med. Chem. Lett. 2009, 19, 3611-3614. DOI: 10.1016/j.bmcl.2009.04.131.
- Kumar, D. P., Reddy V. B. An Efficient, One-Pot, Regioselective Synthesis of 1,4-Diaryl-1H-1,2,3-triazoles Using Click Chemistry. Synthesis, 2010, 1687-1691. DOI: 10.1055/s-0029-1218765.
- Pericherla, K., Khedar, P., Khungar, B., Kumar, A. Click chemistry inspired structural modification of azole antifungal agents to synthesize novel ‘drug like’ molecules. Tetrahedron Lett. 2012, 53, 6761-6764. DOI: 1016/j.tetlet.2012.09.129.
- Nagesh, H. N., Suresh, A., Reddy, M. N., Suresh, N., Subbalakshmia, J., Sekhar, K. V. G. C. Multicomponent cascade reaction: Dual role of copper in thesynthesis of 1,2,3-triazole tethered benzimidazo[1,2-a]quinoline and their photophysical studies. RSC Adv. 2016, 6, 15884-15894. DOI: 1039/C5RA24048D.
- Sindhu, J., Singh, H., Khurana, J. M., Bhardwaj, J. K., Saraf, P., Sharma, C. Synthesis and biological evaluation of some functionalized 1H-1,2,3-triazole tethered pyrazolo[3,4-b]pyridin-6(7H)-ones as antimicrobial and apoptosis inducing agents. Chem. Res. 2016, 25, 1813-1830. DOI: 10.1007/s00044-016-1604-0.
- Fu,, Martínez, A., Wang, C., Ciganda, R., Yate, L., Escobar, A., Moya, S., Fouquet, E., Ruiz, J., Astruc, D. Exposure to air boosts CuAAC reactions catalyzed by PEG-stabilized Cu nanoparticles. Chem. Commun. 2017, 53, 5384-5387. DOI: 10.1039/c7cc02504a.
- Chen, J., Spear, S. K., Huddleston, J. G., Rogers, R. D. Polyethylene glycol and solutions of polyethylene glycol as green reaction media. Green Chem. 2005, 7, 64-82. DOI: 10.1039/B413546F
